

# Out of equilibrium many-body expansion dynamics of strongly interacting bosons

Rhombik Roy[1], Barnali Chakrabarti[1,2*] and Arnaldo Gammal[3]

**1** Department of Physics, Presidency University, 86/1 College Street, Kolkata 700073, India
**2** The Abdus Salam International Center for Theoretical Physics, 34100 Trieste, Italy
**3** Instituto de Física, Universidade de São Paulo, CEP 05508-090, São Paulo, Brazil

⋆ barnali.physics@presiuniv.ac.in

## Abstract

We solve the Schrödinger equation from first principles to investigate the many-body effects in the expansion dynamics of one-dimensional repulsively interacting bosons released from a harmonic trap. We utilize the multiconfigurational time-dependent Hartree method for bosons (MCTDHB) to solve the many-body Schrödinger equation at high level of accuracy. The MCTDHB basis sets are explicitly time-dependent and optimised by variational principle. We probe the expansion dynamics by three key measures; time evolution of one-, two- and three-body densities. We observe when the mean-field theory results to unimodal expansion, the many-body calculation exhibits trimodal expansion dynamics. The many-body features how the initially fragmented bosons independently spreads out with time whereas the mean-field pictures the expansion of the whole cloud. We also present the three different time scale of dynamics of the inner core, outer core and the cloud as a whole. We analyze the key role played by the dynamical fragmentation during expansion. A Strong evidence of the many-body effects is presented in the dynamics of two- and three-body densities which exhibit correlation hole and pronounced delocalization effect.

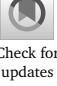

# 1 Introduction

The non-equilibrium dynamics of isolated strongly correlated many-body systems become the most challenging research area both theoretically and experimentally [1,2]. Trapped ultracold atomic gases serve as the ideal testbed to measure the non-equilibrium evolution of the isolated quantum system [3–7]. The dynamical phases acquired by the many-body wave-function for the one-dimensional (1D) gas of impenetrable bosons are studied [8]. A large amount of theoretical research also involves a sudden quench in the Hamiltonian parameter and addresses thermalization and particle transport [9–18]. In inhomogeneous quenches, the most relevant study is the non-equilibrium dynamics of the gas released from a trapping potential. A vast amount of experimental and theoretical works in this direction analyze whether the transport is ballistic or diffusive [16, 19]. Some recent calculations also include the strongly interacting bosons in the Tonks-Girardeau (TG) limit as well as 1D hardcore bosons in optical lattice [2, 20, 21]. In-spite of the availability of extensive work in the field of expansion dynamics of Bose gas, we do not find any work which addresses the many-body effects in the expansion dynamics. This is a significant omission, as it has been predicted that mean-field theory is no longer a good tool for studying expansion dynamics in the strong interaction regime [22]. Furthermore, examining expansion dynamics within a strong interaction limit is significant as the transformative effect of strong interactions on bosons rendering them akin to fermions, a process known as fermionization [23]. This phenomenon leads to a convergence of several characteristics between the two types of gases. Consequently, a compelling avenue of exploration lies in contrasting the expansion behavior of bosons and fermions under strong interaction conditions.

In this article, we explore the expansion dynamics of strongly interacting bosons from a general microscopic quantum many-body perspective. We prepare the initial state at the ground state of few strongly interacting bosons in a harmonic oscillator (HO) potential of finite trap frequency. In the quantum quench, we suddenly turn off the trap and allow the bosons to expand in a large bath. We solve the time-dependent Schrödinger equation (TDSE) by multiconfigurational time-dependent Hartree method for bosons (MCTDHB) at a high level of accuracy and report fully converged many-body wave-function. The many-body wave function is further utilized to understand the transport mechanism from many-body perspective. Our key observations are as follows: (*a*) Utilizing multi-orbital expansion basis in the strongly interactinon limit, we find that the initial state is fragmented whereas, in the mean-field picture, the state is just like a bosonic cloud. (*b*) In the mean-field picture, the whole cloud expands ballistically on release of trap. Whereas in the many-body expansion dynamics, we find independent spreading of $N$ numbers of jets ($N$ is the number of bosons). (*c*) The many-body transport dynamics is characterized by trimodal distribution — expansion of the inner core, outer core, and total cloud expansion. On the other hand, mean-field theory reports only unimodal transport of the whole cloud. (*d*) As MCTDHB utilizes a time adaptive basis, the expansion coefficients, as well as the orbitals, are time-dependent and we can dynamically follow the many-body correlations which is beyond the scope of study in the mean-field theory [24–26] and not addressed earlier in any calculation.

Our present work reports many-body out-of-equilibrium dynamics for a small number of particles $N = 2, 3, 4$ and $5$. In the strong interaction regime, the mean-field state is fragmented, i.e., several orbitals contribute in the dynamics. Fragmentation is the mesoscopic effect that decreases with particle number as $\frac{1}{\sqrt{N}}$ [27–30]. Thus a lower number of particles is preferable to study the beyond mean-field effect. For each case, we find complex nature of transport dynamics. We find that (i) the density splits into number of pieces which is equal to the number of bosons in the trap. Each piece moves distinctly like an independent jet. We observe how the dynamical fragmentation plays the key role in the expansion dynamics. (ii) Trimodal

distribution is the characteristic feature of the few body transport dynamics. (iii) The many-body characteristic features are also observed in the dynamics of two- and three-body density profile. The appearance of a correlation hole along the diagonal of the two-body density due to very strong repulsion mimics the Pauli principle. With time the correlation hole spreads. Additional correlation hole and strong delocalization effects are also observed in the three-body density dynamics.

The paper is organized as follows. In Sec. 2, we introduce the setup and the necessary theory. Sec. 3 explains the basic equation used to measure the different quantities. Sec. 4 explains our numerical results. In Sec. 5, we reach our conclusion.

## 2 Numerical method

The motivation of the present work is to investigate the many-body effect in the expansion dynamics of strongly interacting bosons, released from a trap. Our investigation is based on the dynamical measures of one-body, two-body and three-body densities. The key quantity of this calculation is the computation of exact many-body wave function. We solve the full many-body Schrödinger equation for the interacting $N$-bosons system by MCTDHB which propagates a given many-body state in time. We can solve the time dependent Schrödinger equation at a high level of accuracy. It is demonstrated that MCTDHB can provide exact solutions of time-dependent Schrödinger equation in principle and in practice [31–33].

### 2.1 The many-body wave function

The time dependent Schrödinger equation for $N$ interacting bosons is given by $\hat{H}\psi = i\frac{\partial \psi}{\partial t}$. The total Hamiltonian has the form

$$\hat{H}(x_1, x_2, \ldots x_N) = \sum_{i=1}^{N} \hat{h}(x_i) + \sum_{i<j=1}^{N} \hat{W}(x_i - x_j). \tag{1}$$

The Hamiltonian $\hat{H}$ is in dimensionless unit which is obtained by dividing the dimensionful Hamiltonian by $\frac{\hbar^2}{mL^2}$, $m$ is the mass of the bosons, $L$ is the arbitrary length scale. $\hat{h}(x) = \hat{T}(x) + \hat{V}(x)$ is the one-body Hamiltonian. $\hat{T}(x)$ is the kinetic energy operator and $\hat{V}(x)$ is the external trapping potential. $\hat{W}(x_i - x_j)$ is the two-body interaction. The ansatz for the many-body wave function is the linear combination of time dependent permanents

$$|\psi(t)\rangle = \sum_{\vec{n}} C_{\vec{n}}(t)|\vec{n}; t\rangle. \tag{2}$$

The vector $\vec{n} = (n_1, n_2, \ldots, n_M)$ represents the occupation of the orbitals and $n_1 + n_2 + \ldots + n_M = N$ preserves the total number of particles. In second quantisation representation, the permanents are given as

$$|\vec{n}; t\rangle = \prod_{i=1}^{M} \left( \frac{\left(b_i^\dagger(t)\right)^{n_i}}{\sqrt{n_i!}} \right)|vac\rangle. \tag{3}$$

Distributing $N$ bosons over $M$ orbitals, the number of permanents become $\begin{pmatrix} N+M-1 \\ N \end{pmatrix}$. Thus, if $M \to \infty$, the wave function becomes exact, the set $|n_1, n_2, \ldots, n_M\rangle$ spans the complete $N$-partcle Hilbert space. Although for practical calculation, we restrict the number of orbitals to the desired value requiring the proper convergence in the measure quantities. It is to be noted that both the expansion coefficients $\{C_{\vec{n}}(t)\}$ as well as orbitals $\{\phi_i(x, t)\}_{i=1}^{M}$ that

build up the permanents $|\bar{n}; t\rangle$ are time dependent and fully variationally optimised quantities. Comparing to time-independent basis, as the permanents are time-dependent, a given degree of accuracy is reached with much shorter expansion [34, 35]. We also emphasise that MCTDHB is more accurate than exact diagonalization which uses the finite basis and are not optimised. Whereas in MCTDHB, as we use a time adaptive many-body basis set, it can dynamically follow the building correlation due to inter-particle interaction [24, 36–38] and it has been widely used in different theoretical calculations [25, 39–43]. Thus, to obtain the many-body wave function $|\psi(t)\rangle$, we evaluate the time dependent coefficients $\{C_{\bar{n}}(t)\}$ and orbitals $\{\phi_i(x, t)\}_{i=1}^{M}$. We require the stationarity of the action with respect to the variations of the time dependent coefficients and the time dependent orbitals. It results to a coupled set of equations of motion containing $\{C_{\bar{n}}(t)\}$ and $\phi_i(x, t)$ which are further solved simultaneously [24, 36, 44–46] by recursive MCTDHB (R-MCTDHB) package [32, 33, 47]. It is to be noted that the one particle function $\phi_i(x, t)$ and the coefficient $C_{\bar{n}}(t)$ are variationally optimal with respect to all parameters of the many-body Hamiltonian at any time [48–51]. For getting the ground state, we propagate the MCTDHB equations in imaginary time.

## 2.2 System and chosen parameters

In the present work, we consider few strongly interacting bosons ($N = 2, 3, 4$ and $5$) confined in one-dimensional harmonic oscillator potential and interacting via contact interaction. The many-body Hamiltonian is

$$\hat{H} = \frac{1}{2}\sum_{i=1}^{N}\left(-\frac{\partial^2}{\partial x_i^2} + x_i^2\right) + \sum_{i<j}^{N}\hat{W}(x_i - x_j). \tag{4}$$

For contact interaction,

$$\hat{W}(x_i - x_j) = \lambda\delta(x_i - x_j), \tag{5}$$

where $\lambda$ is the interaction strength determined by the scattering length $a_s$ and the transverse confinement frequency. In this strong interaction regime bosons feel an infinitely repulsive contact interaction and the total energy and density become exactly equal to the energy and density of non-interacting spinless fermions. In the present choice of few boson systems in the harmonic trap, $E_{\lambda\to\infty}^{N} = \frac{N^2}{2}$. We take $\lambda = 25$ for the present calculation. We choose this interaction strength, driven by our intention to investigate expansion dynamics under conditions of strong interaction. In this regime, a deviation between mean-field and many-body results are prominent. Additionally, to observe different modes of expansion, a substantial degree of strong interaction becomes imperative (see sec. 4). At this value of $\lambda$, the number of humps in the one-body density becomes exactly equal to the number of bosons we consider. For the case of four bosons, four humps appear in the one-body density and the energy of the system becomes 8.0. We keep up to $M = 24$ orbitals to guarantee convergence in the measured quantities. Convergence is also discussed in detail in the results section. During quench, we suddenly switch off the trap and allow the bosons to expand in a ring. Our numerical calculation takes place in the range of $x_{\min} = -32$ to $x_{\max} = +32$ with 512 grid points. We perform the simulation using periodic boundary conditions and release the atoms by turning off the trap frequency during the quench. Our primary focus is to examine the expansion dynamics, particularly within a short time. As time progresses, interference patterns emerge at the boundaries. Therefore, we limit our calculation just before this occurs.

# 3 Quantities of interest

For the study of expansion dynamics, we employ the following measures:
(i) The reduced one-body density matrix in coordinate space is defined as

$$\rho^{(1)}(x_1'|x_1;t) = N \int dx_2\, dx_3...dx_N\, \psi^*(x_1', x_2, \ldots, x_N; t)\psi(x_1, x_2, \ldots, x_N; t). \tag{6}$$

The reduced one-body density matrix corresponds to the trace over all particles except one in the complete density operator $|\psi\rangle\langle\psi|$. It is possible to express $\rho^{(1)}(x_1'|x_1;t)$ using a basis representation,

$$\rho^{(1)}(x_1'|x_1;t) = \sum_q \rho_{kq}\phi_k^*(x_1';t)\phi_q(x_1;t) \tag{7}$$

Its diagonal gives the one-body density $\rho(x,t)$ defined as

$$\rho(x;t) = \rho^{(1)}(x_1' = x|x_1 = x;t). \tag{8}$$

(ii) The $p$-th order reduced density matrix in coordinate space is defined by

$$\rho^{(p)}(x_1', \ldots, x_p'|x_1, \ldots, x_p; t) = \frac{N!}{(N-p)!} \int dx_{p+1}...dx_N\, \psi^*(x_1', \ldots, x_p', x_{p+1}, \ldots, x_N; t)$$
$$\times \psi(x_1, \ldots, x_p, x_{p+1} \ldots, x_N; t). \tag{9}$$

Eq. (9) can also be expressed through field operators as

$$\rho^{(p)}(x_1', \ldots, x_p'|x_1, \ldots, x_p; t) = \langle\psi(t)|\hat{\psi}^\dagger(x_1') \ldots \hat{\psi}^\dagger(x_p')\hat{\psi}(x_p) \ldots \hat{\psi}(x_1)|\psi(t)\rangle. \tag{10}$$

The diagonal of $p$-body density can be represented as

$$\rho^{(p)}(x_1, \ldots, x_p; t) = \langle\psi(t)|\hat{\psi}^\dagger(x_1) \ldots \hat{\psi}^\dagger(x_p)\hat{\psi}(x_p) \ldots \hat{\psi}(x_1)|\psi(t)\rangle. \tag{11}$$

It provides the $p$-particle density distribution at time t. In this paper, our focus lies in the computation of both two-body and three-body densities. Eq. (11) is employed for the computation of the two-body density when $p = 2$. When it comes to the calculation of the three-body density, we position the third particle at a designated reference point ($x_3 = x_{\text{ref}}$). Consequently, we are evaluating $\rho^{(3)}(x_1, x_2, x_3 = x_{\text{ref}}; t)$.

# 4 Numerical Results

## 4.1 Expansion dynamics

It is already mentioned that in the limit of $M \to \infty$, as the set of permanents spans the whole Hilbert space, the expansion is exact. So, here we comment on the issue of convergence in our numerical simulation. As the many-body state is fragmented, several natural orbitals have a significant population. So, the choice of natural orbitals is an important issue to address the correct physics. We systematically increase the number of orbitals, and the convergence is assured when the occupation in the highest natural orbital becomes negligible. Additionally, the convergence is quantified when the measured quantities become independent of an

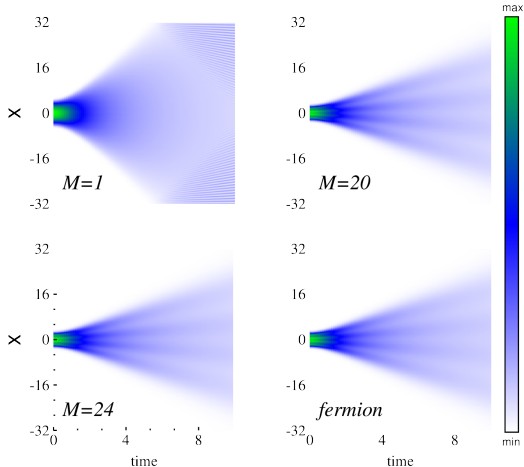

Figure 1: The time evolution of the density profile during the expansion of four strongly interacting bosons initially trapped in a harmonic oscillator (HO) potential and suddenly released. In (a), the expansion dynamics of four bosons are shown in the mean-field limit ($M = 1$). Initially, the bosonic cloud is confined at the center of the HO trap and expands as a whole cloud upon trap release. In (b) and (c), the same expansion dynamics are calculated in the many-body level for $M = 20$ and $M = 24$, respectively. All results are for $\lambda = 25$. The effect of strong repulsion is observed, four jets propagate with time independently. The results obtained with these two different values of $M$ are consistent, indicating the convergence of the many-body calculations using MCTDHB. In (d), the expansion dynamics of four non-interacting spinless fermions are plotted. It ensures that for $\lambda = 25$ the bosons reach fermionized limit. The energy of the four interacting bosons in HO trap becomes 8.0. Figures (b), (c), and (d) are indistinguishable.

increase in the number of orbitals. In Fig.1, we plot the time-evolving one-body density for four strongly interacting bosons. Numerical results are shown for $M = 1$, 20 and 24. $M = 1$ corresponds to the mean-field theory where one single orbital is contributing to the dynamical evolution. Thus, for $M = 1$, the initial state is coherent and not fragmented. With time it remains coherent, just becomes more dilute as it expands in the allowed space. $M = 20$ and $M = 24$ are presented to exhibit the many-body effects and to ensure that the many-body dynamics presented in this article is accurate. It can also be seen that one-body density is indistinguishable for $M = 20$ and 24. However, to ensure convergence, we look for the highest orbital occupation. For $M = 24$, the highest orbital occupation is $\sim 10^{-6}$. So, for the rest of the calculation, we fix $M = 24$. In the same figure, we plot one-body density for $N = 4$ non-interacting fermions which confirm that fermionization limit is reached. For $M = 1$ mean-field calculation, the cloud flows ballistically as a whole. Whereas a strong many-body effect is reconfirmed for $M = 24$ orbitals. Even at $t = 0$, we observe a prominent effect of strong repulsion at the center of the trap; the density is fourfold fragmented. On release from the trap, we observe the propagation of four intense jets. Later, we present that four distinct jets corresponds to four humps in the one-body density when bosons are interacting with infinite repulsive interaction. The expansion of the whole cloud can also be distinguished from the expansion of the humps as their spreading dynamics belong to a different time scale as will be discussed later.

In Fig.2 (a), we plot the one-body density $\rho(x)$ with time for GP mean-field theory. The density profile is Gaussian and represents the whole cloud. With time evolution, the Gaussian cloud only spreads out. This represents that the bosonic cloud expands as a whole. We calcu-

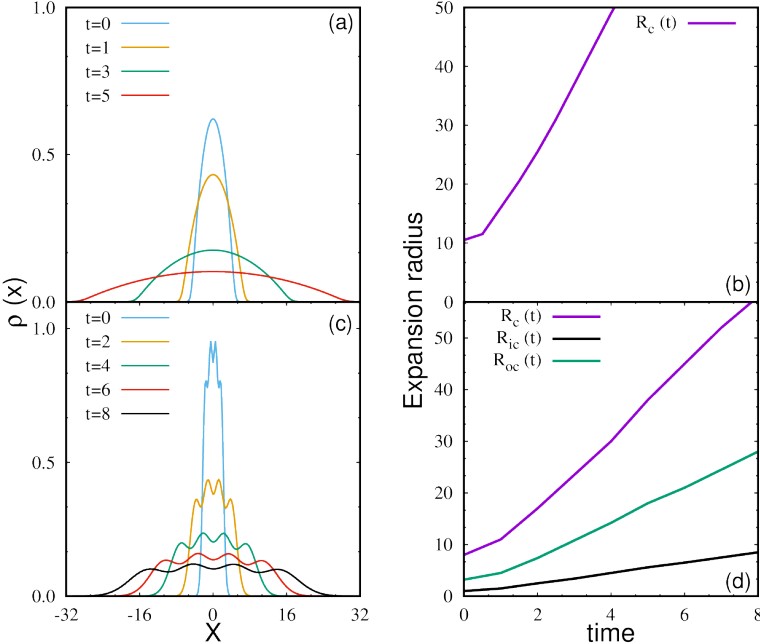

Figure 2: The time evolution of one-body density ($\rho(x)$) during the expansion of strongly interacting bosons with interaction strength $\lambda = 25$ is presented. The results are obtained using both mean-field calculations ($M = 1$) and many-body calculations ($M = 24$). In (a), the one-body density is shown for different times using mean-field calculations, and the expansion is observed to be unimodal. In (b), the expansion radius is plotted against time. In (c), the one-body density is shown for different times using many-body calculations, and the expansion is found to be trimodal, involving the expansion of the two innermost peaks, the expansion radius of the two outermost peaks, and the expansion of the entire cloud. In (d), the expansion radius is shown for three different cases, each with a distinct time scale of expansion.

late the root mean square (RMS) radius as $R_c(t) = \sqrt{\int x^2 \rho(x) dx}$. For GP results, we plot the expansion radius of the whole cloud as a function of time in Fig.2 (b). Fig.2 (c) depicts the many-body results. At $t = 0$, four discrete, closely separated peaks quantify the existence of four strongly repulsive bosons. However, the splitting is localized at the center of the trap due to the effect of the harmonic potential. The inner peaks are prominent, the outermost peaks are less pronounced. After switching off the trap, we monitor the spreading of two innermost peak separations, two outermost peak separations as well as the expansion of the whole cloud as they belong to different time scales. With time, the peaks become flatter and the tail part becomes broader. We define three independent expansion radii. $R_{ic}(t)$, which determines the expansion of two innermost peaks. $R_{oc}(t)$ is the expansion radius of the two outermost peaks. Both $R_{ic}(t)$ and $R_{oc}(t)$ are calculated by the corresponding mutual separation. Whereas core expansion is determined by $R_c(t)$ as described earlier. Fig.2 (d) demonstrates the trimodal expansion in which $R_{ic}(t)$, $R_{oc}(t)$ and $R_c(t)$ are plotted as a function of time. Up to time $t = 2$, we observe deviation from ballistic nature which corresponds to initial expansion. At a later time, all radii increase linearly with time. The trimodal expansion as observed in the many-body expansion dynamics is further described in association with three different expansion velocities ($V_{ex}^{ic}$, $V_{ex}^{oc}$ and $V_{ex}^c$). These are calculated from the corresponding expansion radii as $V_{ex} = \frac{dR(t)}{dt}$. From Fig.2 (d), it is obvious that we can associate three kinds of velocities corresponding to three kinds of expansion dynamics which will be discussed later.

## 4.2 Many-body effect in the expansion dynamics

### Dynamical fragmentation

In Fig.3, we plot the density profile for $N = 2, 3, 4$ and 5 in strong interaction ($\lambda = 25$) limit. In each case, the number of propagating jets is exactly equal to the number of bosons. Initially, the many-body state is fragmented irrespective of the number of bosons. Thus, from the many-body perspective, the expansion dynamics of the fragmented bosons can be characterized by the expansion of the propagating jets. It is in sharp contrast with the mean-field results when the bosonic cloud expands as a 'whole'. However, the expansion velocity increases with the increase in the number of bosons. This is expected as the effective repulsion increases. These

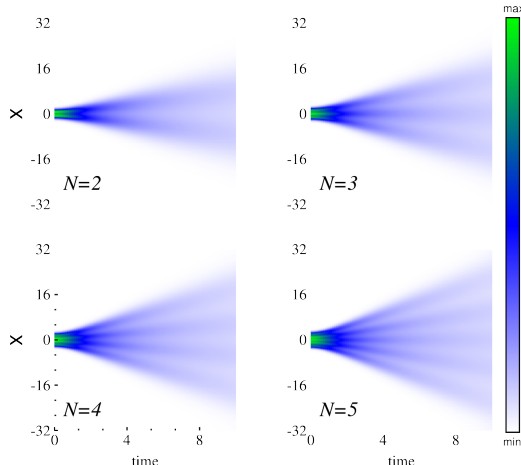

Figure 3: The density profile evolution during the expansion of one-dimensional interacting bosons for varying numbers of particles is shown using a many-body approach. For all cases, we keep $\lambda = 25$ and $M = 24$. Each density profile exhibits a number of independent jets equals to the number of bosons undergoing expansion.

Table 1: Expansion velocities ($V_{ex}^{ic}$, $V_{ex}^{oc}$ and $V_{ex}^{c}$) for different numbers of particles. Velocity increases with increase in the number of particles as the effective interaction strength increases. Quantities are dimensionless.

| $N$ | $V_{ex}^{ic}$ | $V_{ex}^{oc}$ | $V_{ex}^{c}$ |
|---|---|---|---|
| 3 | 1.35 | 2.39 | 6.0 |
| 4 | 2.0 | 3.38 | 6.86 |
| 5 | 2.05 | 4.10 | 7.14 |

observations are concluded from Table.1, where we presented the three kinds of velocity distribution for different numbers of bosons. From the table, it is clear that the expansion velocity is higher for larger number of particles. To analyze the many-body effect, we study the role played by the dynamical fragmentation. It means participation of more than one significantly occupied quantum states during the expansion. Fragmentation is the hallmark of MCTDHB and it plays the key-role for non-equilibrium dynamics of strongly correlated few-body system. In Fig. 4, we plot the evolution of natural occupation in the orbitals having significant contribution. The population in the first natural orbital is always below unity for $N = 2, 3, 4$, and 5 which confirms that fragmentation process is inherent in the many-body dynamics. However,

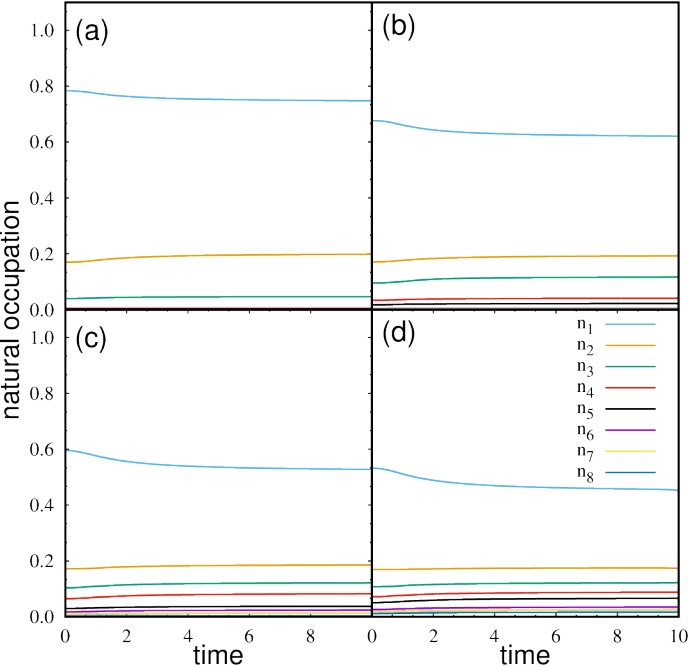

Figure 4: Evolution of occupation in natural orbitals for different number of particles. (a) for $N = 2$, (b) for $N = 3$, (c) for $N = 4$, (d) for $N = 5$. For all cases, we keep $\lambda = 25$ and $M = 24$.

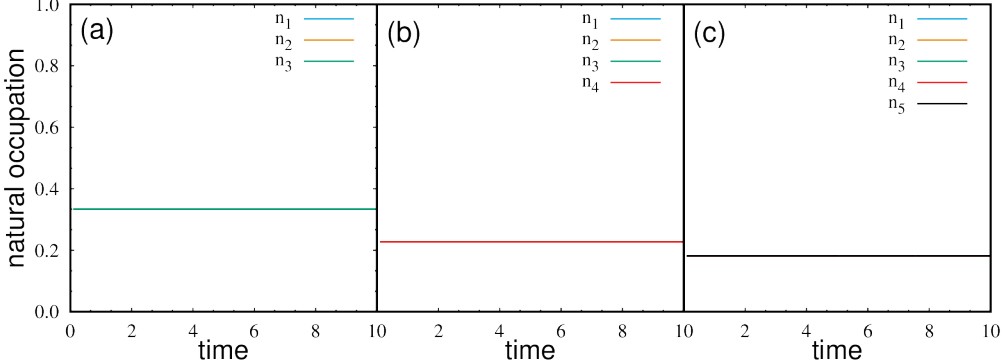

Figure 5: Evolution of occupation in natural orbitals for different number of non-interacting fermions. (a) for $N = 3$; where first three orbitals are equally contributed (each 33%) (b) for $N = 4$; where first four natural orbitals are equally contributed (each 25%) (c) for $N = 5$; where first five orbitals are equally contributed (each 20%). The orbitals overlap and remain constant over time, forming a continuous line.

with an increase in the number of bosons, the degree of fragmentation increases as more orbitals participate in the dynamics. 95% of the population is contributed by the first two natural orbitals for $N = 2$, where as for $N = 3$, first four natural orbitals are required to contribute 95% population. For $N = 4$ and $N = 5$, 95% of the population is contributed by the first six and first eight natural orbitals respectively. In order to elucidate the population dynamics of natural orbitals for fermions, we have undertaken a comparative analysis. We have represented the time evolution of the natural occupation for different number of fermions in Figure 5. The plot reveals a consistent distribution pattern, wherein the fermions are fully fragmented, each

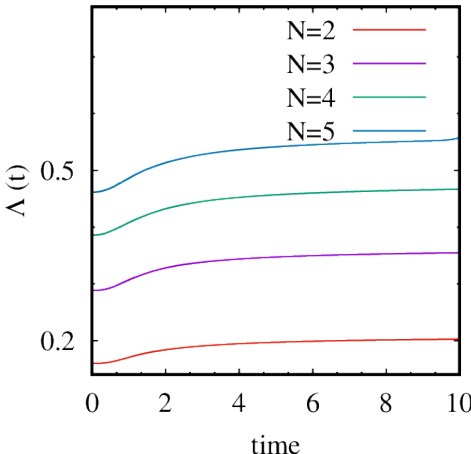

Figure 6: To visualize dynamical fragmentation, $\Lambda(t)$ is plotted as a function of time for different number of particles. See text for further details.

constituting $\frac{1}{N}$ of the total population. To quantify the degree of fragmentation, we define a new measure

$$\Lambda(t) = 1 - n_1(t), \tag{12}$$

for various particle numbers. $\Lambda(t)$ quantifies dynamical fragmentation that is how many other orbitals except the first orbital participate in the dynamics. In Fig. 6, we plot $\Lambda(t)$ as a function of time. At short time (up to $t = 2$), we observe a significant increase in fragmentation, which means the occupation in higher orbitals gradually increases at the cost of occupation in the lowest orbital. This relates the initial expansion as observed in Fig. 2(d). At a longer time, $\Lambda(t)$ maintains almost a steady value implying the occupation in different higher orbitals becomes uniform. It may lead to the conclusion that at a much larger time, the universality of fragmentation would be achieved. Although, the steady value differs with the number of bosons.

### Higher order density dynamics

To elaborate the many-body effects further, we study the dynamics of the density at the two-body and three-body levels. In Fig.7, we plot the two-body density $\rho^{(2)}(x_1, x_2)$ for $N = 3, 4$ and 5 for different time $t = 0.0, 2.0, 5.0$ and 8.0. At $t = 0.0$, we find bosons are localized at the small portion of the system size we consider, but two bosons cannot reside in the same place, rather they are slightly delocalized. The many-body characteristic feature is the appearance of correlation hole across the diagonal when $\rho^{(2)}(x_1, x_2 = x_1) \to 0$. It manifests that the bosons tries to maximize their mutual separation in this strong repulsive interaction. The bright spot along the sub-diagonal or anti-diagonal exhibits how the bosons maximize their mutual distance to minimize the potential energy. With time evolution, the bosons delocalized more, and the width of the correlation hole increases. As the expansion velocity increases with the number of bosons, the delocalization effect is more significant when $N = 5$ compared to $N = 3$ and $N = 4$. Fig. 8 exhibits the many-body effect in the three-body level where we plot time evolution of three-body density for $N = 4$ and 5. For both cases, we fix up the position of the third particle at $x_3 = 0.0$. Thus, an additional correlation hole appears around this fixed point. At this position, the third particle prevents the other bosons to be at this position. For both cases as the bosons expand, the three-body density also exhibits delocalization. It is prominent for $N = 5$ as the effective interaction is more repulsive than $N = 4$.

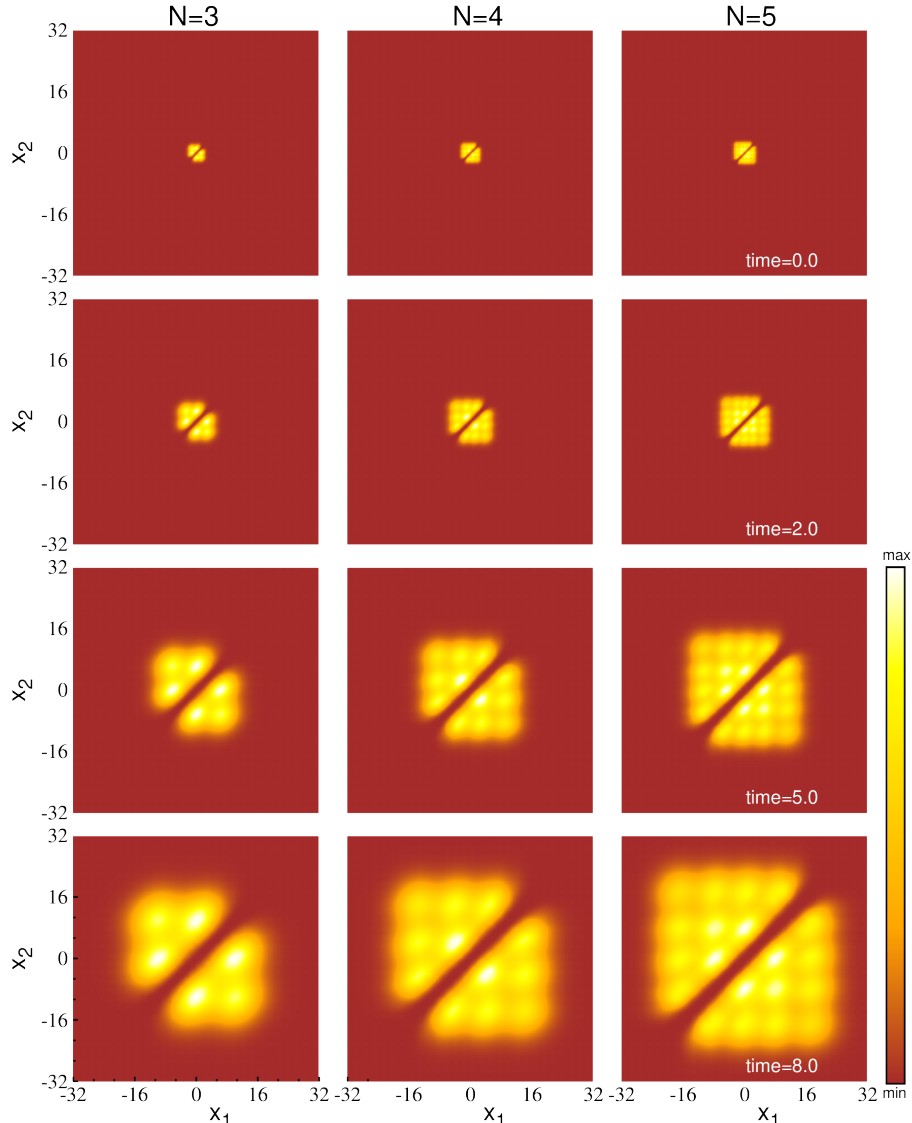

Figure 7: Snapshots of the two-body density $\rho^{(2)}(x_1, x_2)$ for $N = 3$ (left column), $N = 4$ (middle column), and $N = 5$ (right column) in strong interaction regime ($\lambda = 25$) at various stages of expansion. In the strongly interacting regime, the two-body density exhibits a correlation hole along the diagonal ($x_1 = x_2$), which is initially present only in a small portion of the space. As the system expands, these correlations spread out, highlighting the pronounced effect of strong repulsion.

## 5   Conclusion

In this paper, we study the many-body effects in the expansion dynamics of one-dimensional strongly interacting bosons released from a harmonic trap from first principles using the MCT-DHB method. It is well established that both in the non-interacting limit and in the TG limit, the expansion is ballistic. Our motivation is to probe the many-body features in the expansion of density profile in the one-, two- and three-body levels. We observe a very complex expansion dynamics in our present calculation. We conclude that dynamical fragmentation plays a key role to depict the many-body features in expansion dynamics. We monitor three different kinds of expansion: expansion of inner-core, expansion of outer-core, and full cloud

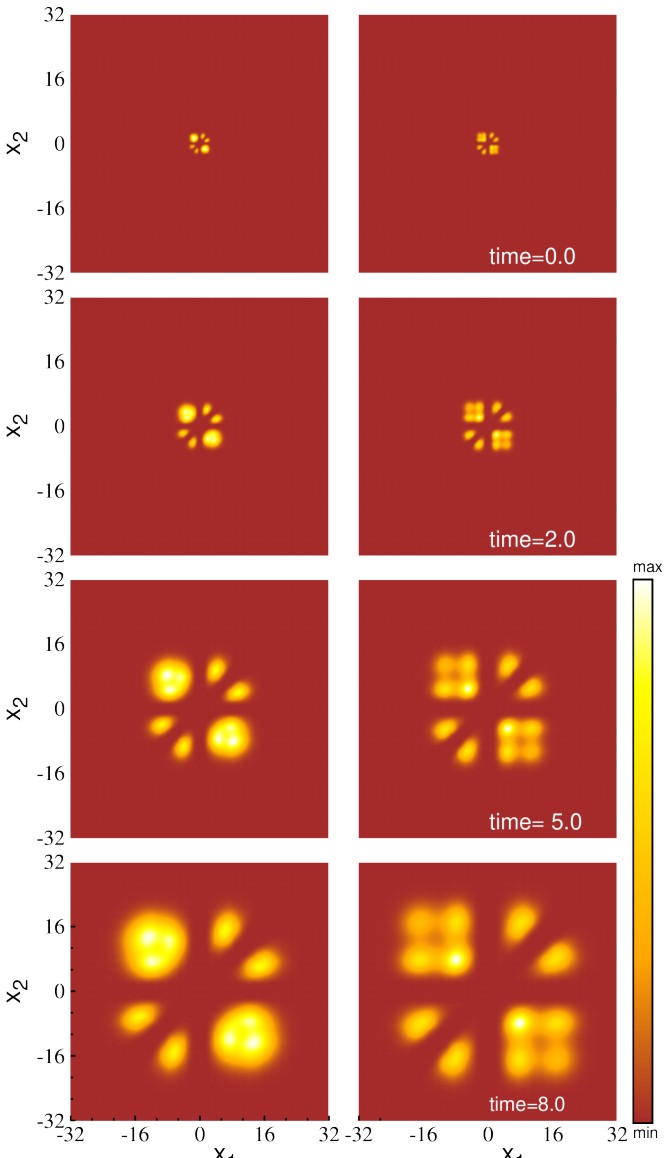

Figure 8: Snapshots of the three-body density $\rho^{(3)}(x_1, x_2, x_3 = 0.0)$ for $N = 4$ (left column) and $N = 5$ (right column) in strong interaction limit ($\lambda = 25$) at different stages of expansion. In the three-body density, with the third boson fixed at $x = 0.0$, additional correlation holes appear surrounding $x = 0.0$, and these correlations expand with time. The other bosons show maximal correlation along the antidiagonal ($x_1 = -x_2$), but the strong effect of repulsion is more prominent for $N = 5$.

expansion. They are associated with three different time scales and the expansion dynamics is trimodal. In contrast, the same expansion dynamics is unimodal in the mean-field theory. In the many-body perspective, the interacting bosons are in a fragmented state when multiple orbitals have a significant contribution. For the fully converged results, we utilize $M = 24$ orbitals in the present calculation. Whereas in the mean-field perspective the condensate is coherent and is described by a single orbital. In the MCTDHB simulation, we can observe significant many-body characteristics during expansion where the mean field predicts the collective ballistic expansion of the bosonic cloud over time. The fragmented bosons spread with time as independent jets. We also report pronounced many-body effects in the dynamics of

higher body density and recognize the formation of a correlation hole and its spreading with time. A strong delocalization effect is also noted in the dynamics of three-body density.

# Acknowledgments

**Funding information** B.C. acknowledges productive discussion and financial support from ICTP, Italy. A.G. also acknowledge the Brazilian agencies Fundação de Amparo à Pesquisa do Estado de São Paulo (FAPESP) [Contract 2016/17612-7] and Conselho Nacional de Desenvolvimento Científico e Tecnológico [Procs. 306920/2018-2].

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
