# Peer review of "Out of equilibrium many-body expansion dynamics of strongly interacting bosons"

_SciPost Physics Core, doi:SciPost Phys. Core 6, 073 (2023)_

## Round 1 · Referee Report · Anonymous · 2023-8-23

Strengths

1- This is a very solid, useful numerical study.

Weaknesses

1- Several important early works are not cited.
2- The value of the coupling constant of choice is too close to the "hard-core" value to be meaningful. It is encouraging that the MCTDHB has been proven to be practically viable, but I would encourage the authors to look at the intermediate couplings, in their next paper.

Report

I am for publishing the paper, provided a few minor changes.

Requested changes

1- An early paper ["Breakdown of Time-Dependent Mean-Field Theory for a One-Dimensional Condensate of Impenetrable Bosons",
M. D. Girardeau and E. M. Wright, Phys. Rev. Lett. 84, 5239 (2000)] constitutes one of the first very explicit demonstrations of the breakdown of the mean-field approximation for strong interactions: it is not cited but it should be.

2- The foundational dynamical fermionization experiment [“Observation of dynamical fermionization”, Joshua M Wilson, Neel Malvania, Yuan Le, Yicheng Zhang, Marcos Rigol, David S Weiss, Science, 367, 1461 (2020)] needs to be cited.

3- The authors need to explain their choice for such a strong coupling ($\lambda = 25.$). I can imagine this was important to test the method, but this needs to be verbalized.

4- Fig. 4, if it is easy to accomplish, I would like to see a hard-core curve here as well.

Minor changes:

5- In "References", "Tonks-Girardeay", "Bose-Einstein", etc need to be capitalized.

6- Figs. 6-7 are missing the value of the coupling constant, in Captions.

---

## Round 2 · Referee Report · Anonymous · 2023-9-19

Report

I am satisfied with the changes made, the manuscript can be published as is.

---

## Round 2 · Author Response

To,
The Editor,

We would like to express our sincere gratitude to the Editor of SciPost Physics Core for their valuable suggestion to revise our manuscript and giving us the opportunity to resubmit it.

We are also deeply appreciative of the meticulous review by the Referee. In accordance with the referee's recommendations, we have diligently revised the manuscript. In this correspondence, we have taken into consideration all of the comments made and have made the necessary corrections accordingly.

After all substantial corrections, we firmly believe that the new physics presented in this article will be of significant interest to the readers of SciPost Physics Core. We earnestly request that you consider processing our manuscript for publication based on the aforementioned revisions.

Thanking you.
R. Roy, B. Chakrabarti, and A. Gammal

---

## Round 2 · List of Changes

The following improvements have been incorporated into our current work:

1. We have appropriately added the necessary citations throughout the manuscript.

2. A discussion on the rationale behind choosing strong interaction has been included in the revised manuscript (found in the last paragraph of page 4).

3. In response to the suggestion of incorporating hard-core bosons, we regret to inform you that we were unable to run the code for hard-core bosons. It is not clear to us which parameters of the hard-core should be chosen. Anyways, it goes beyond our present calculation, and convergence would be a major effort. Instead, we have included a discussion on the fermionic population, (in Figure 5 of new manuscript), and the discussion can be found on page 9.

4. We have revised the references and figure captions as per the recommendations of the referee. We made revisions to both Fig. 7 and Fig. 8, and we also improved the "Quantities of interest" section to enhance its clarity.

---

## Editorial Decision

published